# Gene-Function-Based Clusters Explore Intricate Networks of Gene Expression of Circulating Tumor Cells in Patients with Colorectal Cancer

**DOI:** 10.3390/biomedicines11010145

**Published:** 2023-01-06

**Authors:** Chi-Shuan Huang, Harn-Jing Terng, Yi-Ting Hwang

**Affiliations:** 1Division of Colorectal Surgery, Cheng Hsin General Hospital, Taipei 11220, Taiwan; 2Advpharma, Inc., New Taipei City 22102, Taiwan; 3Department of Statistics, National Taipei University, New Taipei City 22102, Taiwan

**Keywords:** colorectal cancer, circulating tumor cells, intricate network, gene cluster, gene expression

## Abstract

Colorectal cancer (CRC) is a complex disease characterized by dynamically deregulated gene expression and crosstalk between signaling pathways. In this study, a new approach based on gene-function-based clusters was introduced to explore the CRC-associated networks of gene expression. Each cluster contained genes involved in coordinated regulatory activity, such as RAS signaling, the cell cycle process, transcription, or translation. A retrospective case–control study was conducted with the inclusion of 119 patients with histologically confirmed colorectal cancer and 308 controls. The quantitative expression data of 15 genes were obtained from the peripheral blood samples of all participants to investigate cluster–gene and gene–gene interactions. *DUSP6*, *MDM2*, and *EIF2S3* were consistently selected as CRC-associated factors with high significance in all logistic models. *CPEB4* became an insignificant factor only when combined with the clusters for cell cycle processes and for transcription. The *CPEB4/DUSP6* complex was a prerequisite for the significance of *MMD*, whereas *EXT2*, *RNF4*, *ZNF264*, *WEE1*, and *MCM4* were affected by more than two clusters. Intricate networks among *MMD*, RAS signaling factors (*DUSP6*, *GRB2*, and *NF1*), and translation factors (*EIF2S3*, *CPEB4*, and *EXT2*) were also revealed. Our results suggest that limited G1/S transition, uncontrolled DNA replication, and the cap-independent initiation of translation may be dominant and concurrent scenarios in circulating tumor cells derived from colorectal cancer. This gene-function-based cluster approach is simple and useful for revealing intricate CRC-associated gene expression networks. These findings may provide clues to the metastatic mechanisms of circulating tumor cells in patients with colorectal cancer.

## 1. Introduction

Colorectal cancer (CRC) is the third most commonly diagnosed cancer and the second leading cause of cancer-related deaths worldwide, with an estimated 1,880,725 new cases and 915,880 deaths in 2020 [1]. Patients with early-stage CRC are generally easier to treat and have a better prognosis. Advanced imaging techniques have recently been applied for diagnosis, prognosis, and the assessment of treatment response. However, imaging modalities fail to detect dynamic alterations in genetic, histological, and metastatic features induced by environmental influences on tumors within a short time interval [2].

Cancer cells can disseminate from the primary tumor in both the early and late stages of the disease. These rare circulating tumor cells (CTCs) are thought to be highly correlated with distal metastasis, recurrence, and poor clinical outcomes [3,4]. While the optimized isolation method of CTCs that preserves their original status is under development for clinical practice, gene expression in CTCs for diagnosis and prognosis has been intensively studied in recent years [5,6]. Cancer-associated gene signatures obtained from studies of CTC-containing samples could potentially differentiate cancer cases from normal controls and assess the outcomes of patients with CRC [7,8,9,10] and other cancer types, such as lung cancer [11,12,13]. Recent studies have shown that some genes were selected as significant factors in the bivariate analysis but were excluded from the multivariate analysis [7,8,9,10,11,12,13]. Gene–gene interactions presumably existed between the investigated genes, and only stronger effectors or a combination of effectors were selected during the variable selection process. Competitive crosstalk between signaling pathways can be theoretically modeled based on experimental datasets from *Escherichia coli*, yeast, and mammalian cells [14]. In addition, the remarkable regulation of cell cycle processes, such as reversible mitotic arrest, has been reported in dormant cancer cells disseminated in the bloodstream [4,15]. Although different cancer-associated signaling pathways have been intensively studied in cell lines [16,17], little or no information is available regarding crosstalk in gene expression based on clinical data.

In this study, we introduced the concept of a gene-function-based cluster, which is defined as a group of genes with coordinated biological functions or subcellular locations. The expression data of 15 genes obtained from CTC-containing blood samples were used, as viable CTCs seem to be appropriate for studying the cellular mechanisms of metastasis [3,6]. Based on the principle of clinical case–control studies, CRC cases and non-cancer controls were included to construct models to explore the interactions between CRC-associated genes and clusters. Clinically relevant findings could be the basis for capturing the scenario of metastasis-associated regulatory networks and identifying potential therapeutic targets.

## 2. Materials and Methods

### 2.1. Patients, Controls, and Blood Samples

We used a retrospective case–control study to identify the interactions between colorectal cancer-associated genes with the inclusion of 119 cases and 308 non-cancer controls. One hundred and nineteen patients with histologically confirmed colorectal cancer (CRC) were enrolled (2006–2009) in a prospective investigational protocol, which was approved by the Institutional Review Board at Cheng Hsin General Hospital (Taipei, Taiwan).

The non-cancer control group included 308 volunteers who visited our institution for routine health examinations between November 2005 and November 2010. There was no evidence of any clinically detectable cancer diseases at the time of the blood sample collection. The follow-up period of the controls ranged from 4.8 to 9.9 years. Twenty-six controls (8.4%) were censored, and the health statuses of 282 controls were followed up as of September 2015. Of 282 control subjects, 9 (3.2%) were diagnosed with cancer during the follow-up period. The cancer types of these controls included bladder cancer (1), breast cancer (2), ovarian cancer (1), hepatoma (1), urothelial cell carcinoma of the renal pelvis (1), B-cell lymphoma over the bilateral adrenal gland (1), B-cell lymphoma of the stomach (1), and prostate cancer (1).

In addition, among the participants, 111 patients and 227 controls were the same as those used in a previous study by Huang et al. [7], and 308 controls were used in the study by Chian et al. [11]. Peripheral blood samples (6–8 mL) were drawn from participants after obtaining their signed informed consent and then stored at 4 °C until further preparation according to procedures described in a previously published report [7].

### 2.2. Sample Preparation and Relative Quantitative Real-Time PCR

Blood samples (6–8 mL) were collected for the isolation of the mononuclear cell fraction containing tumor cells, followed by total RNA extraction and cDNA synthesis. The relative expression levels (mRNA) of 15 investigated genes in isolated cells of the study sample were measured using quantitative real-time PCR according to a previous report [7]. Pre-designed gene-specific amplification primer sets for the 15 genes and for the reference gene, HPRT1, from Advpharma, Inc. (New Taipei City, Taiwan), nucleotide probes from Universal ProbeLibrary^TM^ (Roche Diagnostics GmbH, Mannheim, Germany), and TaqMan Master Mix (Roche Diagnostics GmbH, Mannheim, Germany) were used for analysis [7,11].

### 2.3. Statistical Analysis

All statistical analyses were performed using SAS (version 9.4; SAS Institute, Cary, NC, USA) with the Logistic procedure (SAS^®^ 9.4 Language Reference: Concepts, 6th Ed., SAS Institute, Cary, NC, USA).

#### 2.3.1. Bivariate Analysis

A chi-square test was used to evaluate the bivariate association between demographics and cancer status, where the age variable was treated as a binary variable, and a cutoff age of 65 years was used. Additionally, the association of each clinico-pathological characteristic between males and females was evaluated using a chi-square test. A logistic regression model was used to evaluate the association between the gene (or cluster) and status (case or control) while controlling for sex and age.

#### 2.3.2. Subsampling and Logistic Regression Analysis of Association with CRC

Because the number of cases (N = 119) was smaller than that of the controls (N = 308), we randomly selected 500 subsamples from the controls with a sample size of 119 to represent heterogeneous populations. A logistic regression model was built for each subsample along with the cases. The derived estimates of the coefficients were summarized as the average of the estimates of the coefficients for 500 samples. Let ${\hat{\beta }}_{j}$ denote the estimate computed from the *j*th sample. Then, the derived estimate is defined as $\frac{1}{500}\sum_{j=1}^{500}{\hat{\beta }}_{j}$. In addition, based on a significance level of 0.05, the percentage of significance of the estimates among 500 models (percent of significance; PS) was computed to evaluate the importance of the gene cluster. In this study, only moderately significant estimates (PS > 50) were analyzed, and an influential gene in the cluster was defined when the PS of the gene in the corresponding cluster was greater than 50. Using such a scheme to find the estimate would yield rather solid and robust associations.

The odds ratios (ORs) and corresponding confidence intervals (CIs) were estimated by exponentiating the derived estimates and the corresponding confidence intervals of the coefficients. The derived standard error (SE) of the derived estimates for the CI includes two parts. The first part is the average of the standard error (wse) of the estimates of the coefficients for the 500 subsamples. The second part is the variation between the samples and is given by the standard deviation (bse) of the estimates of the coefficients between 500 subsamples. The final SE is equal to the square root of the sum of wse^2^ and bse^2^.

### 2.4. Analysis Procedure

A hierarchical analysis procedure was designed to investigate the possible interactions between genes and gene clusters (Figure 1). There were four major steps in this process.

In STEP-1, single-gene modeling was conducted for each investigated gene, and the PS and OR at ground status were obtained.

In STEP-2, (1) 15 investigated genes were grouped into 6 clusters, named LY, TLA, CY, TRf, TRm, and SN, respectively, according to the biological function or subcellular location of each gene as follows: The LY cluster contained *MMD* because its coding protein is involved in the dynamics of lysosomal membranes. The TLA cluster had three translation factors, *EIF2S3*, *EXT2*, and *CPEB4*. Four factors associated with the regulation of the cell cycle process, *MCM4*, *MDM2*, *WEE1*, and *POLDIP2*, were grouped into the CY cluster. The TRf cluster contained two general transcription factors, *ZNF264* and *RNF4*, while the TRm cluster had two factors for the regulation of immune-associated transcription, *IRF4* and *STAT2.* Finally, the SN cluster included three genes involved in RAS signaling, *GRB2*, *NF1*, and *DUSP6*.

(2) Six single-cluster models were constructed to obtain basal cluster-PS (B-PS) and cluster-OR (B-OR) for each cluster-grouped gene, which is written as “gene name/cluster name”.

In STEP-3, a series of multiple-cluster analyses was performed by constructing two-, three-, and four-cluster models. For each model, a primary cluster (or a combination of clusters, called a cluster set) and one cluster listed in square brackets ([ ]) were included in the analysis. Each multiple-cluster model was constructed with the sequential inclusion of clusters in the following order: LY, TLA, CY, TRf, TRm, and SN. For example, the LY/TLA model (model ID: M15) consists of the primary cluster, LY, and the first cluster (TLA) listed in square brackets. Five primary cluster sets were used for the three-cluster models: LY/SN, TLA/SN, LY/TLA, LY/CY, and TLA/CY. Three primary cluster sets were used to construct the four-cluster models: LY/TLA/SN, LY/CY/SN, and TLA/CY/SN.

In STEP-4, gene–gene analysis was performed by constructing two- and three-gene models through the inclusion of two and three genes, respectively, which were grouped in the LY, TLA, and SN clusters.

### 2.5. Analysis of Interactions between Gene Products Using STRING Database

STRING (Search Tool for the Retrieval of Interacting Genes/Proteins) Database version 11.5 was used to analyze protein–protein interaction networks. Two options for the “Search” domain, “Single Protein by Name/Identifier” and “Multiple Proteins by Names/Identifiers”, were input with the names of the 15 investigated genes, using the links https://string-db.org/cgi/input?sessionId=bVqn1xBuFseC&input_page_active_form=single_identifier (accessed on 13 December 2022) and https://string-db.org/cgi/input?sessionId=bVqn1xBuFseC&input_page_active_form=multiple_sequences, respectively (accessed on 17 December 2022).

## 3. Results

### 3.1. Study Sample

This retrospective case–control study was conducted using blood samples from 119 patients with colorectal cancer (CRC) and 308 non-cancer controls. The characteristics of the study sample are presented in Table 1, including the smoking status of participants for both sexes and for two age subgroups (36–65 and 66–89). The proportions of female participants were 54.62% and 43.50% in the case and control groups, respectively. There were statistically significant differences in sex (*p* = 0.039) between CRC cases and non-cancer controls using the chi-square test, but not in age (*p* = 0.644) or smoking status (*p* = 0.157). In addition, there were more female smokers (17.64%) than male smokers (2.52%) in the case group, of whom 12 did not have corresponding information.

Female smokers in the 36–65 age subgroup accounted for approximately 60% of the total female smokers in the case group. Smoking status was not included as a confounding factor in the logistic models because of two considerations: (1) Smoking status was not significantly different between case and control groups. (2) Detailed measurements of cigarette smoking, especially pack-years, were not collected, which might have biased the estimates.

The clinico-pathological characteristics of CRC cases are listed in Table 2. Between males and females, no statistically significant differences as assessed by the chi-square test were identified in the tumor location, stage, TNM classification, or histological grade.

### 3.2. Identification of CRC-Associated Genes by Univariate Analysis

A single-gene model using logistic regression was implemented for each investigated gene to obtain the percent of significance (PS) and odds ratio (OR) for the ground status (Ground-PS and Ground-OR; Figure 1, STEP-1) using logistic regression analysis. As a gene with a PS greater than 50 was defined as a significant CRC-associated factor in the present study, eight genes with significance were identified, seven of which were considered risk genes (Ground-OR > 1) (Table 3) without considering any interactions from other factors.

### 3.3. Interactions between Genes with Coordinated Biological Functions

The fifteen investigated genes were grouped into six clusters, named LY, TLA, CY, TRf, TRm, and SN, respectively, according to their biological functions or subcellular locations, as mentioned in the section on the analysis procedure (Figure 1, STEP-2; Materials and Methods). Six single-cluster models were constructed to yield basal cluster-PS (B-PS) and basal cluster-OR (B-OR) for each cluster-grouped gene (Table 3). The greater the extent of the change in PS (difference between B-PS and Ground-PS), the higher the probability of the existence of an interaction. The results of the single-cluster analysis showed that four cluster-grouped genes (written as “gene name/cluster name”) were possibly influenced by genes grouped in the same cluster, including *EXT2*/TLA, *WEE1*/CY, *STAT2*/TRm, and *GRB2*/SN. In addition, *EXT2*/TLA and *WEE1*/CY were significant CRC-associated factors (PS > 50) in the single-cluster analysis, whereas *STAT2* and *GRB2* lost their significance upon their inclusion in TRm and SN clusters, respectively.

### 3.4. Cluster-Derived Interactions Identified by Two-Cluster Analysis

Multiple-cluster analyses were sequentially performed to identify cluster–gene interactions by constructing two-, three, and four-cluster models (Figure 1, STEP-3). The PS of each cluster-grouped gene in each cluster-based model was obtained with 500 models through subsampling and is presented in Appendix A. If the PS of a cluster-grouped gene in a two- or multiple-cluster model was different from its own basal cluster-PS (B-PS), a cluster–gene interaction possibly existed.

Fifteen two-cluster models were conducted. Five cluster-grouped genes were represented with consistently high significance (PS > 50), and no or negligible effects through inclusion of the second cluster were observed. There were two protective factors (OR < 1), *EIF2S3*/TLA and *EXT2*/TLA, and three risk factors (OR > 1), *CPEB4*/TLA, *MDM2*/CY, and *DUSP6*/SN (Appendix A).

In total, 18 cluster–gene effects were considered valid interactions according to changes in the PS of cluster-grouped genes under any of the following conditions (Table 4): (a) the B-PS of the indicated gene was greater than 50 and then decreased to less than 50; (b) the B-PS of the indicated gene was lower than 50 and then increased to greater than 50; or (c) the B-PS of the indicated gene was greater than 50, and the extent of the change in PS was greater than 5. The cluster–gene interactions are graphically presented in Figure 2, and the extent of the change in PS (∆PS) is shown in Table 4. The results of the two-cluster analysis showed that the PS values of eight cluster-grouped genes were affected by one or multiple clusters, as follows.

The PS of *EXT2*/TLA was positively increased by four other single clusters, namely, CY, TRf, TRm, and SN clusters. *CPEB4*/TLA was influenced by the CY cluster because of its reduced PS. Furthermore, the PS values of two genes grouped in the CY cluster, *MCM4* and *WEE1*, were negatively affected by the TLA or SN clusters. *MCM4*/CY lost its significance when controlling for the TLA cluster, while *WEE1*/CY became an insignificant factor because of the strong negative effect of both TLA and SN clusters. Additionally, *ZNF264*/TRf was negatively affected by four other clusters, with the CY cluster exhibiting the strongest influence. *RNF4*/TRf became a significant factor through an increase in PS when controlling for CY, TRm, or SN clusters. Furthermore, the TRf cluster had a moderate positive effect on the PS of *STAT2*/TRm. Finally, the interaction between the CY cluster and *NF1*/SN was observed.

### 3.5. Complex Cluster–Gene Interactions Identified by Multiple Cluster Analysis

Three- and four-cluster analyses further showed more cluster–gene interactions based on 21 constructed models, and the results are shown in Appendix A. According to their PS, *EIF2S3*/TLA, *EXT2*/TLA, *MDM2*/CY, and *DUSP6*/SN were commonly represented as CRC-associated with high significance (PS = 77.6–100). High expression of *MDM2*/CY and *DUSP6*/SN was significantly associated with a higher risk of colorectal cancer with OR ranges of 5.95–10.62 and 2.14–3.82, respectively. *EIF2S3*/TLA and *EXT2*/TLA were protective factors with OR ranges of 0.17–0.35 and 0.11–0.44, respectively (Appendix A).

Except for *POLDIP2*/CY and *IRF*/TRm, the remaining cluster-grouped genes were influenced by complex interactions according to the variation in their PS for multiple-cluster models. A brief summary of additional findings other than those of the two-cluster analysis is presented as follows.

First, the CY/TRf combination had an intrinsic influence on two genes grouped in the TLA cluster: *CPEB4*/TLA was mostly represented as a significant risk factor (PS = 65.8–100.0), except for the presence of the CY/TRf combination (M88; PS = 38.0). With respect to *EXT2*/TLA, the positive effect derived from the CY cluster (M13; PS = 99.8) or TRf cluster (M9; PS = 90.0) disappeared in the TLA/CY/TRf model (M88; PS = 77.6).

Second, three cluster-grouped genes became significant CRC-associated factors in the presence of certain cluster combinations: TLA/SN set for *MMD*/LY (M28, M17, M20, and M21), LY/TLA/SN set for *STAT2*/TRm (M21), and LY/TLA set for *GRB2*/SN (M28, M20, and M21; Figure 3a).

Third, the CY cluster combined with the LY/TRf or TLA/TRf set had an enhanced positive effect on *NF1*/SN (M41 versus M5 or M85 versus M14; Figure 3b). However, this CY-derived effect vanished in the absence of the TRf cluster, for example, in TLA/CY/SN-based models (M23 versus M14 and M17 versus M5; Figure 3b).

Fourth, with respect to two factors (*MCM4* and *WEE1*) in the CY cluster, the findings of the TLA-derived negative effect on their PS were mostly similar to those obtained from the two-cluster analysis. *MCM4*/CY and *WEE1*/CY lost significance (PS < 50) upon the inclusion of the TLA cluster in the three- and four-cluster models (M14, M16, M17, M85, M87, M88, and M105; Figure 4). However, the addition of the TRf or TRm cluster to the LY/CY/SN set slightly compensated for the SN-derived negative effect on *MCM4*/CY (M41 versus M5 and M43 versus M5; Figure 4a).

Fifth, *ZNF264*/TRf and *RNF4*/TRf were oppositely affected by CY- or LY/CY-containing models in the three-cluster analysis (M40, M88, and M41; Figure 5). The CY-derived negative effect on *ZNF264*/TRf was partially reversed by the coexistence of TLA/SN (M85); however, the LY/TLA/SN set suppressed the PS of *ZNF264*/TRf (M20). In addition, the TLA cluster eliminated the SN-derived positive influence on the PS of *RNF4*/TRf (M6 vs. M11).

### 3.6. Intricate Networks between Genes Involved in Translational Regulation and RAS Signaling

*MMD*, the only gene in the LY cluster, became a significant factor for LY/TLA/SN-containing models. According to this finding, we constructed 11 two-gene models and 15 three-gene models (STEP-4; Figure 1) based on the combination of seven genes, which were grouped into LY, TLA, and SN clusters. The association of each gene in the multiple-gene model with CRC was examined by logistic regression analysis and represented by PS (Table 5) and compared with its own Ground-PS (Table 3). Three genes, *CPEB4*, *GRB2*, and *DUSP6*, were considered risk factors for colorectal cancer disease (OR > 1; Table 6), and EIF2S3 and EXT2 were considered protective genes.

The complex interactions between seven genes are schematically presented in Figure 6. The analysis results showed that three genes, *EIF2S3*, *CPEB4*, and DUSP6, were consistently represented as CRC-associated factors with high significance (PS > 98.0; Table 5) in all multiple-gene models. Four genes were affected by other genes or combinations as follows: *EXT2*, *EIF2S3*, and *MMD* could interfere with the *CPEB4*-derived negative effect on *GRB2*, whose PS increased from 23.4 (M57) to 99.8 (M59), 79.8 (M58), and 52.4 (M33), respectively. Moreover, the PS of *NF1* was positively affected by *EIF2S3* (M53) or *CPEB4* (M48), but not by the *EIF2S3*/*CPEB4* combination (M49). However, this *EIF2S3–NF1* interaction was impeded by the addition of *MMD* (M45 versus M53). In addition, the significance of *EXT2* was increased in the presence of *GRB2* (M56), *DUSP6* (M62), *CPEB4*/DUSP6 (M65), *CPEB4*/*GRB2* (M59), or *CPEB4*/*NF1* (M50). Finally, *MMD* was identified as a significant CRC-associated factor in the *MMD*/*CPEB4*/*DUSP6* model (M36).

### 3.7. Interactions between Gene Products Using STRING Database

The search results for interaction networks of “Single Protein” based on the STRING database showed that no interactions were the same as those identified in this study (Appendix A). Furthermore, regarding the networks of multiple proteins, five protein–protein interactions were identified (Figure 7). Known interactions were marked for IRF4–STAT2 and WEE1–GRB2, and co-expression was identified for NF1–MDM2, whereas two interactions, WEE1–MDM2 and GRB2–DUSP6, were indicated as “textmining” (co-mentioned in PubMed Abstracts). In addition, no association was found for other proteins.

## 4. Discussion

Cancer cells are characterized by the deregulation of cell signaling [16,17], which is suggested to activate relevant oncogenic pathways and facilitate invasion ability and disease dissemination [18]. In this study, the novel approach was the gene-function-based cluster; each includes genes involved in coordinated biological functions or subcellular locations. We have six clusters: SN for RAS signaling, CY for the cell cycle process, TRf for general transcription, TRm for immune-associated transcription, TLA for translation, and LY for lysosomal membrane proteins. The utility of gene-function-based clusters for the analysis of the complex regulation of colorectal cancer-associated gene expression in circulating tumor cells is disclosed.

Single-cluster models uncover interactions between genes involved in different steps of coordinated cellular processes: The SN model reveals interactions between the downstream ERK1/2 inactivator *DUSP6* and the upstream regulator *GRB2* of RAS/ERK signaling [19,20], whereas this interaction is represented as “textmining” according to the STRING database. The CY model shows that the G2 checkpoint kinase *WEE1* [21] is strongly influenced by the combination of DNA replication-associated molecules, including *MCM4*, *MDM2*, and *POLDIP2*. Furthermore, *EIF2S3* and *CPEB4* together had noticeable effects on *EXT2* using the TLA model. In addition, the existence of the *IRF4–STAT2* interaction is revealed by the TRm model, and this agrees with the search results obtained from the STRING database. Based on the current knowledge, *IRF4* is involved in *STAT3*-oncogenic signaling [22], whereas *STAT2* is reported to be associated with *STAT1* and interferon regulatory factor 9 (IRF9) to form IFN-stimulated gene factor 3 (ISGF3) [23].

The complexities of cluster–gene interactions indicate gene regulations across different pathways in circulating tumor cells. First, as *EXT2*/TLA is positively affected by four different clusters, *EXT2* presumably plays a central role in the invasive character of colorectal cancer with respect to extracellular matrix (ECM) assembly [24,25,26]. Second, the significance of *ZNF264*/TRf was suppressed by four different clusters, whereas the CY cluster exerted the strongest effect. This finding is concordant with a report on the involvement of *ZNF264*-based transcriptional activity during the cell cycle process, especially replication stress and genomic instability in cancer [27]. Thirdly, the notable increase in the significance of *RNF4*/TRf by the CY cluster may indicate that *RNF4* likely participates in the DNA damage response and the maintenance of chromosomal integrity during the cell cycle through the SUMO-targeted ubiquitin ligase (STUbL) pathway [28]. Based on this assumption, the SN-derived positive influence on the significance of *RNF4*/TRf is acceptable, since the impactful factor ERK1/2 inactivator *DUSP6*/SN [19,20] may drive the reduction in the mitogenic signal. The fourth and the fifth cluster-grouped genes with varying significance were *MCM4*/CY and *WEE1*/CY, in which strong and intricate modulations derived from SN and TLA clusters were observed. It is presumed that factors involved in RAS signaling and translation closely coordinate or counteract each other to regulate the cell cycle process.

Interaction networks between LY, TLA, and SN clusters are supposedly associated with the regulation of the cell cycle process. For instance, *GRB2*/SN only represents a significant factor for the coexistence of the LY/TLA combination but without the CY cluster. In contrast, the CY cluster was crucial for the significance of *NF1*/SN. Thus, *NF1* might be required for spindle organization and chromosome segregation in circulating tumor-derived cells, as in neuron cells [29], rather than for the inactivation of the Ras protein [30]. In addition, the complexity of modulations can be also observed for two transcription factors grouped in the TRf cluster. The TLA-derived negative effect on the significance of *ZNF264*/TRf was partially reversed in the TLA/SN combination model, whereas the TLA-derived effect was enhanced by the addition of the LY cluster. Moreover, the SN-derived positive effect on the significance of *RNF4*/TRf was completely suppressed by the TLA/SN combination. Finally, our results suggest that *CPEB4* may be involved not only in the translational regulation of mitosis as reported [31] but also in DNA replication, since the CY cluster consistently had a negative effect on the significance of *CPEB4*/TLA. Furthermore, *CPEB4*/TLA unexpectedly became an insignificant factor in the presence of the CY/TRf combination.

Two-gene and three-gene models confirmed intricate interactions between *MMD* and factors involved in RAS signaling and translation. The presence of the *CPEB4*/*DUSP6* combination is a prerequisite for the significance of *MMD*, which supports the reported five-gene model for colorectal cancer [7,8]. Lower *MMD* expression may indicate a low amount of active ERK1/2 if *MMD* is regulated by LPS stimulation in macrophages, as reported by Liu et al. [32]. Based on this assumption, the combination of low *MMD* expression and *DUSP6* overexpression significantly impedes the transduction of mitogenic signals. It is presumed that some circulating tumor cells are likely to exhibit reversible G0-G1 arrest and have high metastatic potential, as reported for dormant cancer cells in the circulation [4,15]. Moreover, limited mitotic activity has been reported to be favorable for the increased malignancy of endometrial cancer stem-like cells [33] and chemotherapy resistance in cancer [34,35,36]. In addition, two factors of RAS signaling pathways, *GRB2* and *DUSP6*, may be involved in the invasiveness of circulating tumor cells through intricate interactions with the suppressor gene *EXT2*. Finally, we verified the presumption that translation factors may participate in the regulation of cell proliferation via the MAPK/ERK signaling pathway, since *EIF2S3* and *CPEB4* showed strong but opposite effects on *NF1*. Our results for *EIF2S3* expression in colorectal cancer-derived cells are in accordance with findings in acute myeloid leukemia [37].

## 5. Conclusions

Overall, *DUSP6*, *MDM2*, and *EIF2S3* were consistently selected as significant factors associated with colorectal cancer in all logistic models and were not modulated by any other genes or clusters. These results suggest that limited G1/S transition, uncontrolled DNA replication, and the cap-independent initiation of translation may be dominant and concurrent scenarios in circulating tumor cells from colorectal cancer. The primary strength of this study is the approach based on gene-function-based clusters for the identification of complex interactions between factors involved in mitogenic signaling, cell cycle processes, transcription, and translation in cells with high metastatic potential. Moreover, the cluster–gene interactions identified in this study are novel. Most of the interactions between proteins encoded by the 15 investigated genes have been neither published nor included in the STRING database. In addition, these results provide clues for the varied gene-specific associations with colorectal cancer in univariate and multivariate analyses. Thus, the approach based on gene-function-based clusters is expected to be useful for the identification of rational gene signatures for clinical diagnostic and prognostic utilities, as well as for the validation of drug targets.

Our study had several limitations. First, the findings of cluster–gene and gene–gene interactions might not be associated with colorectal cancer, since patients with other cancer types were not included in the investigation. Second, the subjects in the control group were only confirmed to be cancer-free, without knowing other health conditions, such as inflammatory bowel diseases; thus, interference with the identified interactions in this study could not be excluded. Third, the appropriateness of the grouping of genes is uncertain because the cellular functions of some proteins encoded by the investigated genes are not fully understood. For instance, proteins exert oncogenic activity only after translocation from the cytosol to the nucleus. Furthermore, we lack the knowledge of proteins that may function as double-edged swords. Therefore, the grouping of gene clusters in this study may provide a partial scope for crosstalk between regulatory pathways in cancer cells. Fourth, only age and sex were controlled for in the logistic models, whereas other confounding variables, such as measurements of cigarette smoking, body mass index (BMI), and other lifestyle risk factors, were not fully collected during the inclusion period. Fifth, the small sample size and the proportion of patients with different clinical stages of the disease may have influenced the results.

Future research should include different grouping principles of genes to identify more novel interactions. Moreover, how these factors, when located in different subcellular compartments, interact with each other requires further investigation. In addition, the examination of the specificity of cluster–gene and gene–gene interactions for colorectal cancer is required through investigations of other cancer types. Multiple drug targets could be potentially conceived to develop advanced therapeutic agents in precision medicine.

It was concluded that combined cluster-based and gene–gene analyses can be used to explore the crosstalk between cellular activities and rationally represent parallel scenarios in colorectal cancer-derived cells. Multiple gene-based signatures can provide a better overview of the characteristics of circulating tumor cells isolated from patients with colorectal cancer and further dynamic and personalized information for prognoses and therapeutic responses.

## Figures and Tables

**Figure 1 biomedicines-11-00145-f001:**
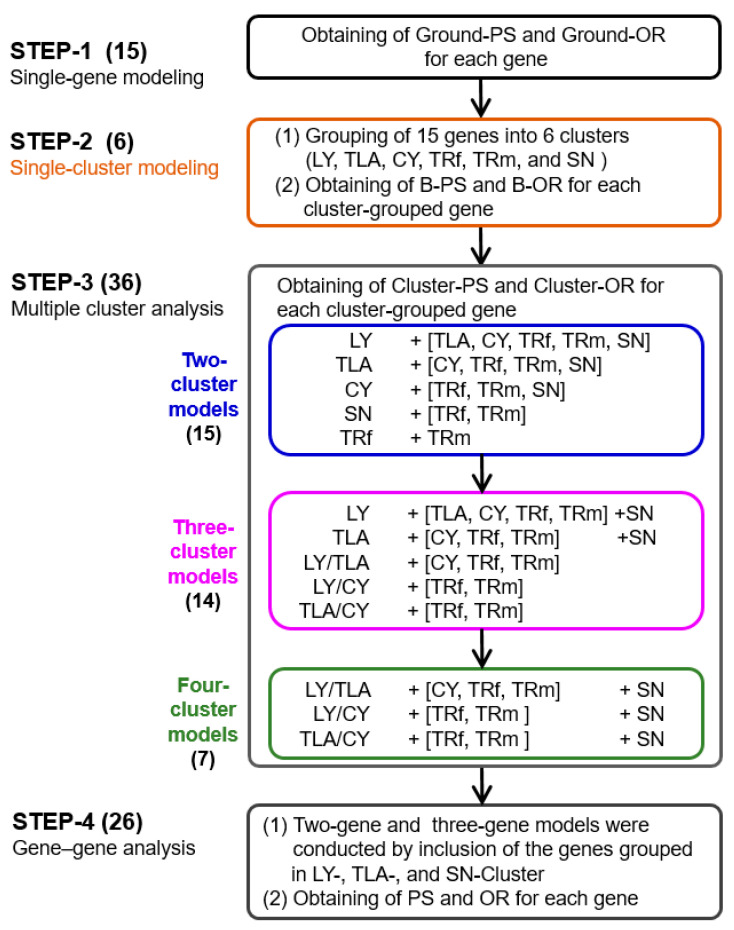
Analysis procedure for cluster–gene and gene–gene interactions. Four major steps were performed, as described in detail in the Section 2. The total number of constructed models for corresponding steps or substeps is indicated in parentheses. Abbreviations: PS, percent of significance; OR, odds ratio; Ground-PS, PS for single-gene model; Ground-OR, OR for single-gene model; B-PS, basal cluster-PS or PS for single-cluster model; B-OR, basal cluster-OR or OR for single-cluster model.

**Figure 2 biomedicines-11-00145-f002:**
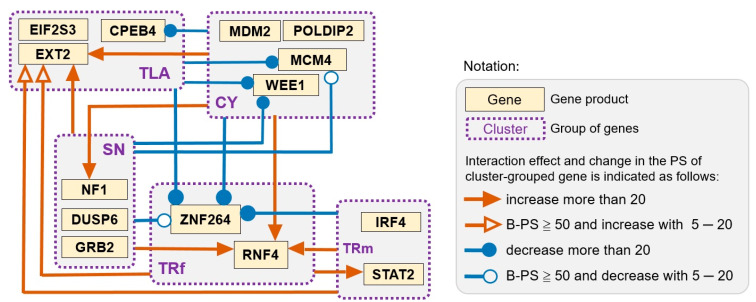
Schematic presentation of valid cluster–gene interactions identified by two-cluster analysis. String derived from the edges of a cluster indicates the existence of cluster-derived influence on the pointed gene grouped in another cluster. The magnitude of the change in the percent of significance (PS) of the pointed gene from its basal PS (B-PS) is shown with an arrow (increase) or dot (decrease) next to the pointed gene. The LY cluster is not shown because of the absence of interaction with other clusters.

**Figure 3 biomedicines-11-00145-f003:**
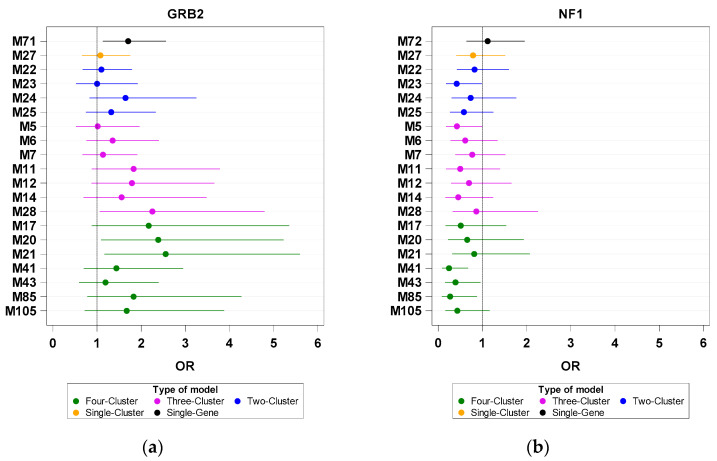
Odds ratios (ORs) of investigated genes based on single-gene, single-cluster, and multiple-cluster analyses. The X- and Y-axes are the OR and Model ID, respectively. (**a**) *GRB2* and (**b**) *NF1*.

**Figure 4 biomedicines-11-00145-f004:**
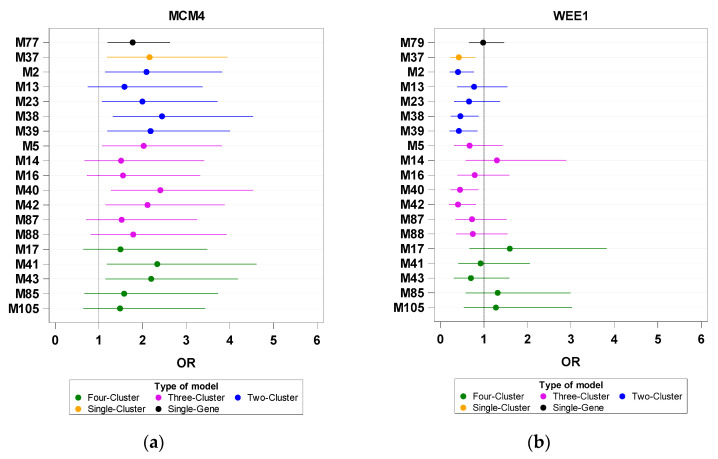
Odds ratios (ORs) of investigated genes based on single-gene, single-cluster, and multiple-cluster analyses. The X- and Y-axes are the OR and Model ID, respectively. (**a**) *MCM4* and (**b**) *WEE1*.

**Figure 5 biomedicines-11-00145-f005:**
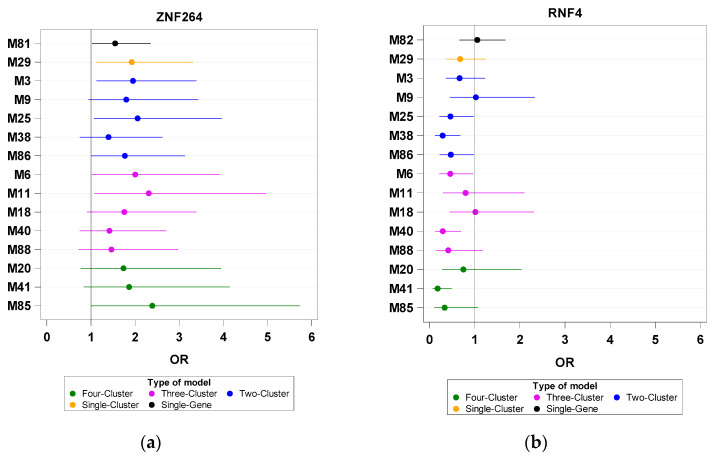
Odds ratios (ORs) of investigated genes based on single-gene, single-cluster, and multiple-cluster analyses. The X- and Y-axes are the OR and Model ID, respectively. (**a**) *ZNF264* and (**b**) *RNF4*.

**Figure 6 biomedicines-11-00145-f006:**
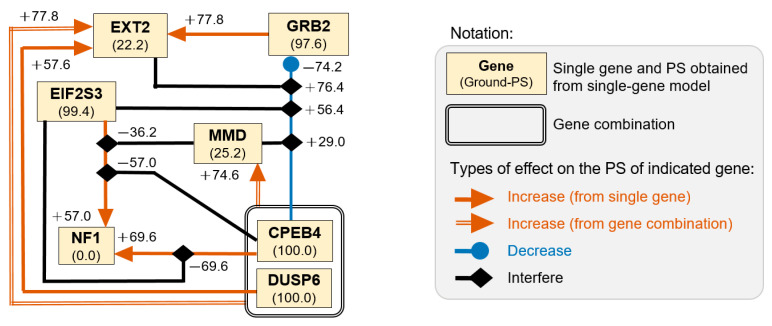
Schematic presentation of interactions between seven genes according to logistic analysis data (Table 5). The extent of change in percent of significance (PS) is indicated next to the notation for the interaction effect. Ground-PS, PS obtained from single-gene model (Table 3).

**Figure 7 biomedicines-11-00145-f007:**
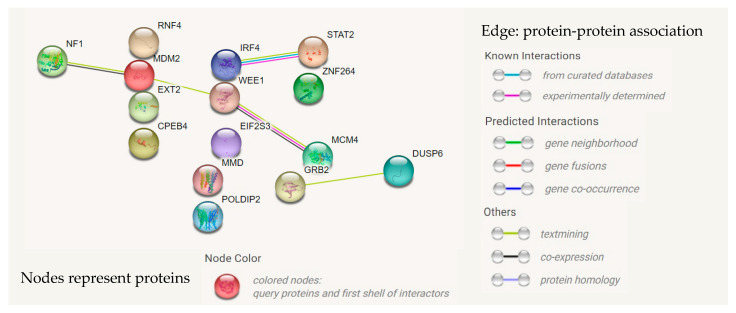
Network analysis of 15 proteins encoded by 15 investigated genes using STRING database version 11.5. The option “Multiple Proteins by Names/Identifier” of “Search” domain was used. Visualization of interactions and notations is according to the output of database.

**Table 1 biomedicines-11-00145-t001:** Basic demographics of the study sample (N = 427).

		CRC Case (N = 119)				Control (N = 308)			
		Male		Female		Male		Female	
Smoking Status		n	%	n	%	n	%	n	%
36–65 years old									
	Smoker	1	0.84	13	10.92	32	10.39	7	2.27
	Non-smoker	17	14.29	17	14.29	34	11.04	54	17.53
	Missing	2	1.68	2	1.68	0	0.00	0	0.00
66–89 years old									
	Smoker	2	1.68	8	6.72	33	10.72	19	6.17
	Non-smoker	26	21.85	23	19.33	75	24.35	54	17.53
	Missing	6	5.04	2	1.68	0	0.00	0	0.00

Abbreviation: CRC, colorectal cancer.

**Table 2 biomedicines-11-00145-t002:** Clinico-pathological characteristics of case group (N = 119).

		Total		Male		Female		*p* ^§^
Characteristics		n	%	n	%	n	%	
Tumor Location								0.63
	Colon	72	60.50	35	29.41	37	31.09	
	Rectum	41	34.45	17	14.29	24	20.17	
	RSJ ^1^	6	5.04	2	1.68	4	3.36	
Pathological Stage								0.72
	0–I	33	27.73	14	11.76	19	15.97	
	II	28	23.53	15	12.61	13	10.92	
	III	37	31.09	17	14.29	20	16.81	
	IV	21	17.65	8	6.72	13	10.92	
TNM Classification								
	pT							0.90
	pTis	6	5.04	3	2.52	3	2.52	
	pT1	12	10.08	5	4.20	7	5.88	
	pT2	19	15.97	7	5.88	12	10.08	
	pT3	65	54.62	30	25.21	35	29.41	
	pT4	17	14.29	9	7.56	8	6.72	
	pN							0.71
	pN0	65	54.62	31	26.05	34	28.57	
	pN1	30	25.21	12	10.08	18	15.13	
	pN2	23	19.33	11	9.24	12	10.08	
	pNX	1	0.84	0	0.00	1	0.84	
	M							0.46
	M0	98	82.35	46	38.66	52	43.70	
	M1	21	17.65	8	6.72	13	10.92	
Histological Grade								0.33
	G1	5	4.20	2	1.68	3	2.52	
	G2	85	71.43	43	36.13	42	35.29	
	G3	9	7.56	3	2.52	6	5.04	
	Missing	20	16.81	6	5.04	14	11.76	

**^§^** The *p* value was obtained from the chi-square test. ^1^ Rectosigmoid junction.

**Table 3 biomedicines-11-00145-t003:** Logistic analysis for single-gene and single-cluster models.

Gene	Single-Gene Analysis					Single-Cluster Analysis				
ID	Ground-PS	Ground-OR	LB	UB	Model ID	B-PS	B-OR	LB	UB	Model ID
*MMD*	25.2	1.22	0.94	1.59	M1	25.2	1.22	0.94	1.59	M1
*EIF2S3*	99.4	0.42	0.22	0.79	M74	100.0	0.33	0.16	0.66	M8
*EXT2*	22.2	0.72	0.47	1.12	M75	76.0	0.57	0.33	0.98	M8
*CPEB4*	100.0	2.43	1.70	3.49	M76	100.0	3.07	2.03	4.63	M8
*MCM4*	100.0	1.77	1.20	2.63	M77	97.2	2.16	1.18	3.94	M37
*MDM2*	100.0	5.08	2.79	9.26	M78	100.0	5.89	3.00	11.55	M37
*WEE1*	0.0	0.98	0.66	1.46	M79	99.2	0.42	0.22	0.80	M37
*POLDIP2*	21.0	1.34	0.89	2.01	M80	0.0	0.93	0.53	1.62	M37
*ZNF264*	73.8	1.55	1.02	2.35	M81	92.8	1.92	1.12	3.31	M29
*RNF4*	0.0	1.06	0.66	1.68	M82	15.4	0.68	0.37	1.24	M29
*IRF4*	8.6	1.25	0.88	1.78	M83	0.0	1.06	0.70	1.59	M30
*STAT2*	81.4	1.52	1.03	2.25	M84	39.0	1.48	0.95	2.31	M30
*GRB2*	97.6	1.71	1.14	2.56	M71	0.0	1.08	0.66	1.75	M27
*NF1*	0.0	1.12	0.64	1.95	M72	1.2	0.79	0.41	1.52	M27
*DUSP6*	100.0	2.64	1.73	4.01	M83	100.0	2.62	1.62	4.22	M27

Abbreviations: PS, percent of significance; Ground-PS, PS for single-gene model; B-PS, basal cluster-PS, PS for single-cluster model; OR, odds ratio; Ground-OR, OR for single-gene model; B-OR, basal cluster-OR, OR for single-cluster model; LB, lower bound of 95% confidence interval; UB, upper bound of 95% confidence interval.

**Table 4 biomedicines-11-00145-t004:** Cluster–gene interactions summarized from two-cluster analysis.

					Cluster					
	TLA		CY		TRf		TRm		SN	
Cluster-Grouped Gene	∆PS	Model ID	∆PS	Model ID	∆PS	Model ID	∆PS	Model ID	∆PS	Model ID
*EXT2*/TLA			23.8	M13	14.0	M9	18.6	M10	21.4	M24
*CPEB4*/TLA			−31.0	M13						
*MCM4*/CY	−80.2	M13							−9.8	M23
*WEE1*/CY	−98.2	M13							−87.6	M23
*ZNF264*/TRf	−38.0	M9	−90.6	M38			−23.0	M86	−8.8	M25
*RNF4*/TRf			83.4	M38			57.8	M86	60.8	M25
*STAT2*/TRm					26.2	M86				
*NF1*/SN			70.4	M23						

Abbreviations: PS, percent of significance; ∆PS, changes in PS of affected cluster-grouped genes (written as “Gene/Cluster”), calculated according to Appendix A.

**Table 5 biomedicines-11-00145-t005:** Logistic analysis for gene–gene interactions. Percent of significance (PS) of each gene for each two-gene or three-gene model is listed.

Model ID	Combination	MMD	EIF2S3	EXT2	CPEB4	GRB2	NF1	DUSP6
Two-gene models								
M44	*MMD*/*CPEB4*	3.0			100.0			
M48	*CPEB4*/*NF1*				100.0		69.6	
M53	*EIF2S3*/*NF1*		99.8				57.0	
M54	*EXT2*/*NF1*			28.2			0.2	
M55	*EIF2S3*/*GRB2*		100.0			100.0		
M56	*EXT2*/*GRB2*			100.0		100.0		
M57	*CPEB4*/*GRB2*				100.0	23.4		
M61	*EIF2S3*/*DUSP6*		100.0					100.0
M62	*EXT2*/*DUSP6*			79.8				100.0
M63	*CPEB4*/*DUSP6*				100.0			100.0
M68	*MMD*/*DUSP6*	0.4						100.0
Three-gene models								
M31	*MMD*/*EIF2S3*/*GRB2*	7.8	100.0			99.0		
M32	*MMD*/*EXT2*/*GRB2*	0.2		100.0		100.0		
M33	*MMD*/*CPEB4*/*GRB2*	20.8			100.0	52.4		
M34	*MMD*/*EIF2S3*/*DUSP6*	0.0	100.0					100.0
M35	*MMD*/*EXT2*/*DUSP6*	0.2		77.2				100.0
M36	*MMD*/*CPEB4*/*DUSP6*	99.8			100.0			100.0
M45	*MMD*/*EIF2S3*/*NF1*	42.2	100.0				20.8	
M46	*MMD*/*EXT2*/*NF1*	25.2		31.0			0.0	
M47	*MMD*/*CPEB4*/*NF1*	0.6			100.0		62.2	
M49	*EIF2S3*/*CPEB4*/*NF1*		100.0		100.0		0.0	
M50	*EXT2*/*CPEB4*/*NF1*			94.6	100.0		49.8	
M58	*EIF2S3*/*CPEB4*/*GRB2*		100.0		100.0	79.8		
M59	*EXT2*/*CPEB4*/*GRB2*			100.0	100.0	99.8		
M64	*EIF2S3*/*CPEB4*/*DUSP6*		100.0		100.0			100.0
M65	*EXT2*/*CPEB4*/*DUSP6*			100.0	100.0			100.0

**Table 6 biomedicines-11-00145-t006:** Logistic analysis for gene–gene interactions. Odds ratios (ORs) of each gene for each two-gene or three-gene model are listed.

Model ID	Combination	*MMD*	*EIF2S3*	*EXT2*	*CPEB4*	*GRB2*	*NF1*	*DUSP6*
Two-gene models								
M44	*MMD*/*CPEB4*	0.83			2.73			
M48	*CPEB4*/*NF1*				2.89		0.48	
M53	*EIF2S3*/*NF1*		0.31				1.90	
M54	*EXT2*/*NF1*			0.71			1.21	
M55	*EIF2S3*/*GRB2*		0.32			2.05		
M56	*EXT2*/*GRB2*			0.43		2.47		
M57	*CPEB4*/*GRB2*				2.29	1.39		
M61	*EIF2S3*/*DUSP6*		0.29					3.06
M62	*EXT2*/*DUSP6*			0.57				2.88
M63	*CPEB4*/*DUSP6*				2.09			2.22
M68	*MMD*/*DUSP6*	0.89						2.83
Three-gene models								
M31	*MMD*/*EIF2S3*/*GRB2*	1.20	0.31			1.90		
M32	*MMD*/*EXT2*/*GRB2*	1.10		0.43		2.36		
M33	*MMD*/*CPEB4*/*GRB2*	0.76			2.66	1.52		
M34	*MMD*/*EIF2S3*/*DUSP6*	0.95	0.29					3.15
M35	*MMD*/*EXT2*/*DUSP6*	0.90		0.58				3.05
M36	*MMD*/*CPEB4*/*DUSP6*	0.58			2.76			2.89
M45	*MMD*/*EIF2S3*/*NF1*	1.28	0.30				1.68	
M46	*MMD*/*EXT2*/*NF1*	1.23		0.70			1.07	
M47	*MMD*/*CPEB4*/*NF1*	0.86			3.15		0.49	
M49	*EIF2S3*/*CPEB4*/*NF1*		0.33		2.88		0.82	
M50	*EXT2*/*CPEB4*/*NF1*			0.51	3.28		0.51	
M58	*EIF2S3*/*CPEB4*/*GRB2*		0.26		2.54	1.65		
M59	*EXT2*/*CPEB4*/*GRB2*			0.26	2.77	2.46		
M64	*EIF2S3*/*CPEB4*/*DUSP6*		0.24		2.32			2.55
M65	*EXT2*/*CPEB4*/*DUSP6*			0.38	2.50			2.55

## Data Availability

The data presented in this study are available in Appendix A.

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
