# Peer review of "Gene-Function-Based Clusters Explore Intricate Networks of Gene Expression of Circulating Tumor Cells in Patients with Colorectal Cancer"

_biomedicines, 2023, doi:10.3390/biomedicines11010145_

Round 1

Reviewer 1 Report

Very insightful and engaging work you introduced!

Here is my recommendation for you;

Is there any other study discussing gene-to-gene interaction using a statistical method identical to you in other malignancies?

Author Response

The authors gratefully thank to the Referee for the constructive comments and recommendations which definitely help to improve the readability and quality of the paper.

Point 1: Is there any other study discussing gene-to-gene interaction using a statistical method identical to you in other malignancies?

Response 1:

We introduce a new approach combining the gene function-based clusters and the unique inclusion principle of cluster(s) for logistic regression analysis to investigate the correlation of expressions of gene-cluster(s) with colorectal cancer. This approach has not been reported in any study for malignancies. 

Reviewer 2 Report

The manuscript by Huang and co-workers addresses an interesting and pertinent topic, given the need to gain further insight into the networks of gene expression underlying colorectal cancer pathogenesis and metastization, contributing to improve its diagnosis and prognosis. Study limitations are appropriately acknowledged. The conclusions are adequate.

Despite its current flaws, which I detail below, I believe that this paper fits the scope of Biomedicines and that it could be considered for publication, once the following concerns are met.

- I suggest that the whole manuscript is revised for formatting, since several aspects of the Biomedicines template have not been observed (e.g., the text font in some sections, the author line formatting, bibliographical reference formatting, figure and table (non-)incorporation into the Results section).

- Title: please replace “expressions” by “expression”.

- Line 66: please remove the period between “Escherichia” and “coli”.

- Line 91: please either remove the “a” before “routine health examinations” or replace “examinations” by “examination”.

- Although some ideas are implicitly conveyed, I suggest opportunities for future research to be briefly, but explicitly, stated at the end of the Discussion section.

Author Response

The authors gratefully thank to the Referee for the constructive comments and recommendations which definitely help to improve the readability and quality of the paper. All the comments are addressed accordingly and have been incorporated to the revised manuscript. Detailed responses to the comments and recommendations are as follows.

Point 1:  I suggest that the whole manuscript is revised for formatting, since several aspects of the Biomedicines template have not been observed (e.g., the text font in some sections, the author line formatting, bibliographical reference formatting, figure and table (non-)incorporation into the Results section).

Response 1: Format of the revised manuscript has been prepared according to the Biomedicines Microsoft Word template file.

Point 2: Title: please replace “expressions” by “expression”.

Response 2: Correction has been accordingly performed in the revised manuscript.

Point 3: Line 66: please remove the period between “Escherichia” and “coli”.

Response 3: Correction has been accordingly performed in the revised manuscript (Line 60).

Point 4: Line 91: please either remove the “a” before “routine health examinations” or replace “examinations” by “examination”.

Response 4: Correction has been accordingly performed in the revised manuscript (Line 83).

Point 5: Although some ideas are implicitly conveyed, I suggest opportunities for future research to be briefly, but explicitly, stated at the end of the Discussion section.

Response 5: We added as the third paragraph in the Section of 5. Conclusion, Line 468-473, as follows.

Future research should include different grouping principle of genes to identify more novel interactions. Moreover, how these factors locating in different subcellular compartments interact with each other, requires further investigation. In addition, the examination of the specificity of cluster—gene and gene—gene interactions for colorectal cancer is required by investigations of other cancer types. Multiple drug targets could be potentially the concept to develop advanced therapeutic agents in precision medicine.

Reviewer 3 Report

Specific comments to the authors

The authors Chi-Shuan Huang et al. of the submitted manuscript “Gene function-based clusters explore intricate networks of gene expressions in patients with colorectal cancer” studied the CRC-associated networks of gene expressions via gene function-based cluster methods in detail. Therefore, the authors used a case-control study with 119 cases with CRC and 308 controls by isolation of the mononuclear cell fraction of peripheral blood samples and investigation with relative quantitative real-time PCR.

In summary, based on their investigations of a case-control study authors demonstrated that limited G1/S transition, uncontrolled DNA replication, and cap-independent initiation of translation are leading mechanism in circulating tumor cells derived from colorectal cancer. Therefore, the authors concluded that gene function-based cluster approach is a statistical technique for (i) revealing the intricate CRC-associated gene expression networks and for (ii) investigating insights in metastatic mechanisms of circulating tumor cells in patients with colorectal cancer.

Overall, the manuscript give an interesting approach to cluster gene expression data of circulation tumor cells of CRC. The manuscript is sometimes not easy read and to follow. Therefore, the present must be essentially changed. The methods are mostly well described. Although the results and discussion are not clearly presented, the authors (see specific comments) must perform some minor to major changes to improve the manuscript. In conclusion, the presented data are interesting. After incorporating the mentioned specific comments (see below) the manuscript has the potency to be accepted.

Specific comments

Title: The used title “Gene function-based clusters explore intricate networks of gene expressions in patients with colorectal cancer” does not reflect that circulating tumor cells are investigated by the authors. Please change adequately.

Abstract: Please add the type of study (case-control, prospective) in more detail.

Material and Methods:

Please explain why six clusters are used for the “step 2” statistical procedure. The used/applied abbreviations are not clear (“LY”, “TLA”,”CY”, TRf”, “TRm” and “SN”) and should be clearly described by the authors.

Results: Overall, the result section is hard to read and to follow. Therefore, the authors should clarify the cluster analysis by visualization of the findings. Furthermore, the definitive readouts of the investigations are not clear. The descriptive statics of the gene expression data should be summarized in a separate table.

# Table 1: Please add more clinico-pathological characteristics (regarding grading and TNM-staging) of the cases. Furthermore, the authors should give more details of the analyzed blood samples (such as DNA, reads, copy numbers…).

# Table 2: The used term “G-PS” is not standard and should be changed and transferred to “classical” p-values.

# Figure 2 and 3: The authors should perform KEGG analysis for the gene-cluster and gene–gene interactions. Furthermore, these findings should be included in the figure 2 and 3.

Discussion: The discussion repeats descriptively the findings of the results. The authors should evaluate/discuss the possible biological “ranking” of the found gene clusters in the chosen clinical/experimental setting. Furthermore, the authors should discuss the findings with published studies. The main concerns to the fact, that the authors investigated gene expression of circulating tumor cells of patients CRC, whereby this connection is not mentioned in the discussion. Furthermore, any kind of relation to the tumor stage is missing, too. Finally, how could the interesting findings transferred from a theoretical to a practical view (like clinical setting (diagnostics/therapeutics))? Please discuss in short.

Reviewer 4 Report

Research article revealed major complex interaction between genes and clusters. Various cancer associated signaling pathways are associated with each other through genes function cluster.

1. Result part of the paper is difficult to follow and understand because of lack of figures.

2. Figure no. 2 and 3 are very complex figures and difficult to understand. Please simplify the figure

3. Line 143 what is the meaning of LY here?

4.Line 156 what is CN, please correct this in paper

Author Response

The authors gratefully thank to the Referee for the constructive comments and recommendations which definitely help to improve the readability and quality of the paper. All the comments are addressed accordingly and have been incorporated to the revised manuscript. Detailed responses to the comments and recommendations are as follows.

Point 1: Result part of the paper is difficult to follow and understand because of lack of figures.

Response 1: Results in general:

(1) We introduce a new approach combining the gene function-based clusters and the unique inclusion principle of cluster(s) for logistic regression analysis to investigate the correlation of expressions of gene-cluster(s) with colorectal cancer.

(2) Percent of significance (PS) was obtained from 500 samplings instead of one single analysis.

To take into account the heterogeneous population in controls, 500 subsamples of size 119 were randomly selected from the controls. Using the case and each subsample of controls, the logistic model was used to assess the association between the investigated genes or clusters and CRC. The estimation procedure was repeated 500 times and 500 estimates were derived. To understand the significance of the association from 500 models, we counted the number of significances and computed the percent of significances from 500 models. Thus, the Ground-PS was formed from the classical p values.

(3) Before cluster analysis:

In order to identify the “real” cluster-derived impact on the investigated genes, we started the analysis with single-gene and single-cluster models, with which we had the background information about the CRC-association of each single gene and each cluster-grouped gene, respectively. 

(4) Cluster analysis:

We firstly conducted models for all pairwise combinations of clusters (15 two-cluster models) and then selectively conducted multiple-cluster models for exploration of impacts derived from particular gene groups (clusters) as so-called “primary cluster”. For example, 14 multiple-cluster models contains SN-cluster (factors involved in RAS signaling) could provide information for the impact derived from RAS signaling pathway on other regulations of gene expressions.

(5) Statistical data of all cluster-based analysis are listed in Supplementary Table S1, while some selective data of cluster analysis (Table 4; Figure 3-5) and multiple-gene models (Table 5-6 and Figure 6) are inserted in the revised manuscript.

(6) Visualization of cluster analysis results:

Since each cluster contains 2-4 genes (except for LY-cluster), the cluster-derived effect is resulted from the gene combination and might rationally reflect the multiple scenarios in the “real” living cells. The presentation of networks of cluster-derived interactions has been simplified in Figure 2. The corresponding data (magnitude of change in PS) for understanding of Figure 2 are listed in Table 4.

Point 2: Figure no. 2 and 3 are very complex figures and difficult to understand. Please simplify the figure

Response 2: Both figures have been simplified.

(1) Figure 2 presents intricate cluster-derived interactions. The string from the edges of cluster indicates the existence of influence on the pointed gene grouped in another cluster. Arrow or dot next to the gene (in box) in Figure 2 indicates the impact (increase or decrease) on the significance level (in percent of significance, PS in manuscript) of the pointed gene (in cluster). Since each cluster contains 2-4 genes (except for LY-cluster), the cluster-derived effect is resulted from the gene combination and might rationally reflect the scenarios in real cell. The corresponding data (magnitude of change in PS) for understanding of Figure 2 are listed in Table 4.

(2) Figure 3 of primary submitted manuscript has been changed to Figure 6 for revised version. It presents the complex networks of interactions between seven genes, including MMD and factors involved in RAS signaling and translational regulation.

Point 3: Line 143 what is the meaning of LY here?

Response 3: LY-cluster contains the only gene, MMD, because of its coding protein involved in the dynamics of lysosomal membranes and its biological function is not clear. In Section 2.4. STEP-2, the information for the abbreviated cluster name has been added as follows (Line 144-152):

15 investigated genes were grouped into six clusters, named as LY, TLA, CY, TRf, TRm, and SN, respectively, according to biological function or subcellular location of each gene as published: LY-cluster contains MMD because of its coding protein involved in the dynamics of lysosomal membranes. TLA-cluster had three translational factors, EIF2S3, EXT2, and CPEB4. Four factors associated with regulation of the cell cycle process, MCM4, MDM2, WEE1, and POLDIP2, were grouped into CY-cluster. TRf-cluster contained two general transcriptional factors, ZNF264 and RNF4, while TRm-cluster had two factors for regulation of immune-associated transcription, IRF4 and STAT2. Finally, SN-cluster included three genes involved in the RAS signaling, GRB2, NF1, and DUSP6.

Point 4: Line 156 what is CN, please correct this in paper

Response 4: The typographical error has been corrected in revised manuscript (Line 162).

Round 2

Reviewer 3 Report

Specific comments to the authors

In the revised version of the manuscript, the authors could address previous mentioned concerns in a very adequate and convincing manner. Therefore, the revised manuscript “Gene function-based clusters explore intricate networks of gene expression of circulating tumor cells in patients with colorectal cancer" should be accepted.